# Epidemiological description of Marburg virus disease outbreak in Kagera region, Northwestern Tanzania

Vida Mmbaga[1], George Mrema[1,2]*, Danstan Ngenzi[1], Welema Magoge[1], Emmanuel Mwakapasa[1], Frank Jacob[1], Hamza Matimba[3], Medard Beyanga[3], Angela Samweli[4], Michael Kiremeji[4], Mary Kitambi[4], Erasto Sylvanus[4], Ernest Kyungu[5], Gerald Manase[5], Joseph Hokororo[6], Christer Kanyankole[7], Martin Rwabilimbo[7], Issessanda Kaniki[7], George Kauki[8], Maria Ezekiely Kelly[8], William Mwengee[8], Gabriel Ayeni[8], Faraja Msemwa[8], Grace Saguti[8], George S. Mgomella[9], Kokuhabwa Mukurasi[9], Marcelina Mponela[9], Eliakimu Kapyolo[10], Jonathan Mcharo[10], Mary Mayige[10], Wangeci Gatei[9], Ishata Conteh[11], Peter Mala[12], Mahesh Swaminathan[9], Pius Horumpende[13], Paschal Ruggajo[13], Grace Magembe[14], Zabulon Yoti[8], Elias Kwesi[4], Tumaini Nagu[15]

1 Epidemiology and Disease Control Section, Ministry of Health, Dodoma, Tanzania, 2 Tanzania Field Epidemiology and Laboratory Training Program, Ministry of Health, Dar es Salaam, Tanzania, 3 National Public Health Laboratory, Ministry of Health, Dar es Salaam, Tanzania, 4 Emergency Preparedness and Response Unit, Ministry of Health, Dodoma, Tanzania, 5 President Office Regional Administrative Local Government, Dodoma, Tanzania, 6 Health Quality Assurance Unit, Ministry of Health, Dodoma, Tanzania, 7 Regional Health Management Team, Kagera, Tanzania, 8 World Health Organization, Dar es Salaam, Tanzania, 9 US Centers for Disease Control and Prevention, Dar es Salaam, Tanzania, 10 National Institute for Medical Research, Dar es Salaam, Tanzania, 11 World Health Organization, AFRO, Brazzaville, Congo, 12 World Health Organization, HQ, Geneva, Switzerland, 13 Directorate of Curative Services, Ministry of Health, Dodoma, Tanzania, 14 Office of the Permanent Secretary, Ministry of Health, Dodoma, Tanzania, 15 Office of the Government Chief Medical Officer, Ministry of Health, Dodoma, Tanzania

* drgeorgemrema@gmail.com

**Data Availability Statement:** All relevant data are within the manuscript and its Supporting Information files.

## Abstract

### Introduction

In March 2023, a Marburg Virus Disease (MVD) outbreak was declared in Kagera region, Northwestern Tanzania. This was the first MVD outbreak in the country. We describe the epidemiological characteristics of MVD cases and contacts.

### Methods

The Ministry of Health activated an outbreak response team. Outbreak investigation methods were applied to cases identified through MVD standard case definitions and confirmed through reverse-transcriptase polymerase chain reaction (RT PCR). All identified case contacts were added into the contact listing form and followed up in-person daily for any signs or symptoms for 21 days. Data collected from various forms was managed and analyzed using Excel and QGIS software for mapping.

**Funding:** The author(s) received no specific funding for this work.

**Competing interests:** The authors have declared that no competing interests exist.

## Results

A total of nine MVD cases were reported with eight laboratory-confirmed and one probable. Two of the reported cases were frontline healthcare workers and seven were family related members. Cases were children and adults between 1–59 years of age with a median age of 34 years. Six were males. Six cases died equivalent to a case fatality rate (CFR) of 66.7%. A total of 212 individuals were identified as contacts and two (2) became cases. The outbreak was localized in two geo-administrative wards (Maruku and Kanyangereko) of Bukoba District Council.

## Conclusion

Transmission during this outbreak occurred among family members and healthcare workers who provided care to the cases. The delay in detection aggravated the spread and possibly the consequent fatality but once confirmed the swift response stemmed further transmission containing the disease at the epicenter wards. The outbreak lasted for 72 days but as the origin is still unknown, further research is required to explore the source of this outbreak.

## Introduction

Marburg virus disease (MVD) is a severe and highly fatal viral infection caused by the Marburg virus. MVD is known for its severe hemorrhagic fever symptoms, including high fever, headaches, muscle pain, and bleeding from various parts of the body [1]. Within *Filoviridae* family, there are two distinct genera, *Orthomarburgvirus* and *Orthoebolavirus* responsible for causing highly contagious viral hemorrhagic fevers. The Orthomarburgvirus genus contains a unique species, *Orthomarburgvirus marburgense*, with two members Marburg virus (MARV) and Ravn virus (RAVV) [2]. Studies have shown that Fruit bats (*Rousettus aegyptiacus*) are the natural hosts and reservoirs for the MARV [3–6].

MARV was discovered in 1967 after infected monkeys were brought to laboratories in Marburg (Germany) and Belgrade (former Yugoslavia) from Uganda causing laboratory-based MVD outbreaks in the respective cities [7, 8]. There have been outbreaks in several African countries, including Kenya (1987), the Democratic Republic of Congo (1998), Angola (2005), Uganda (2007, 2012, 2014, 2017), and Equatorial Guinea (2023) [9–16]. In 2008, the virus was also detected in the United States of America (USA) and the Netherlands after travelers returned from Uganda and had visited a popular fruit bat cave in a national park [1, 17].

On September 20, 2022, an outbreak of Sudan Virus Disease (SVD) caused by *Orthoebolavirus sudanense* was declared by the government of Uganda in Mubende district Western Uganda. The outbreak, which recorded 164 cases (142 confirmed and 22 probable) and 77 deaths (55 confirmed and 22 probable), was declared over in January 2023 [18]. This outbreak spread to nine districts, including Masaka City in southern Uganda, which is approximately 90 km (55 miles) from Mutukula town in Kagera region, Tanzania. SVD and MVD share common symptoms and are both transmitted through contact [19]. The outbreak of SVD in Uganda led the Tanzania Ministry of Health (MOH) and its partners to activate preparedness and operational readiness in the Kagera region, which was vital in responding to MVD. Extensive training in events-based surveillance (EBS) was carried out at the community level, and a mobile laboratory equipped with viral hemorrhagic fever (VHF) testing capabilities was

strategically placed in the region. Additionally, a simulation exercise for responding to SVD was also carried out in Kagera region in February 2023.

On March 21, 2023, the first MVD outbreak was declared in Kagera region, in the north-western part of Tanzania, following laboratory confirmation from samples collected from a cluster of illnesses and deaths in two wards in Bukoba district between March 14 and 16, 2023. The outbreak lasted for 72 days and was declared over by the Tanzania MOH on June 1, 2023. This article describes the epidemiological characteristics of cases and contacts of MVD outbreaks to inform future outbreak prevention, preparedness, and response measures.

## Methods

### Outbreak area

The outbreak occurred in Kagera region located in Northwestern Tanzania. Kagera region borders Rwanda to the west and Burundi to the southwest. To the north, Kagera region shares a border with Uganda, where previous MVD outbreaks have been reported [11, 12, 14, 15].

It has an area of 1,654 square kilometers, 28 wards, and a population of 2, 989, 299 people according to the 2022 Tanzania Population and Housing Census [20]. The residents engage in various economic activities such as cultivation, animal husbandry, fishing, and commerce.

### Formulation of outbreak response team

Following the official declaration of the outbreak, the Tanzania Ministry of Health (MOH) created a dual-level response team at the national and regional levels. The response teams were composed of multidisciplinary experts led by incident managers. The national level team was responsible for providing leadership, technical support, formulating guidelines and standard operating procedures, as well as mobilizing resources while the regional response teams led in all response operations. Teams were organized into eight pillars: coordination, case management, surveillance, laboratory and sample management, logistics, psychosocial support, risk communication community engagement (RCCE), water sanitation and hygiene (WASH), and point of entry (PoE). Specialists from national and international partners joined the respective response pillars at the regional level. Residents from the Field Epidemiology and Laboratory Training Program (FELTP) were also deployed throughout the outbreak period supporting different pillars.

**Case findings.** Case finding was conducted by dedicated Rapid Response Teams (RRTs), consisting of surveillance officers, laboratory experts, and clinicians. The RRTs were deployed to investigate verified MVD alerts from health facilities or the community rapidly completing preliminary investigations. The working MVD case definition was adopted from the World Health Organization (WHO)'s VHF case definitions and included suspect probable and confirmed cases [21].

A suspected case was defined as any person who lived in or had traveled to Kagera Region in the last 21 days (three weeks) and had the following features (i) three or more symptoms consistent with MVD: Fever of 38.0˚C (or 37.5˚C axillary) or higher, headache, vomiting, nausea, diarrhea, abdominal pain, loss of appetite, general body weakness, aching muscles or joints, unexplained bleeding, cough, rash, difficulty swallowing, difficulty in breathing, hiccups and convulsions, (ii) presenting with unexplained bleeding (in vomitus, stool or diarrhea; bleeding from nose, eyes, mouth and/or skin) and (iii) any sudden and unexplained death. A probable case was any person (dead or alive) meeting the suspect case definition, who had an epidemiological link with a confirmed MVD case (dead or alive) but was not tested or did not have a laboratory confirmation. A confirmed case was any suspected or probable case dead or alive with a positive laboratory result for Marburg virus by reverse-transcriptase polymerase

chain reaction (RT-PCR). These MVD Standardized Case Definitions (SCD) were distributed in all health facilities and PoE.

After the RRT's verification and confirmation that a case met the SCD, a Case Investigation Form (CIF) was completed which was subsequently transformed to populate the line list which contains key information about each case, and a sample was collected and sent to an established mobile laboratory in the outbreak area for RT-PCR testing. The assay used to confirm the outbreak was Altona Diagnostics RealStar® Filovirus Screen RT-PCR capable of detecting various Orthoebolaviruses and distinguishing Marburg virus. The samples were tested at the mobile laboratory and transported to the National Public Health Laboratory (NPHL) in Dar es Salaam for confirmation.

To enhance case finding, an alert desk and additional hotline numbers were established at the regional level, complementing the pre-existing full-time, toll-free number (199) that the MOH utilizes for receiving alerts from the public. To facilitate community reporting of alerts, community alert cards containing symptoms of MVD were introduced and distributed to the community.

**Contact tracing and follow-up.** Contact tracing was done by a team comprising surveillance officers, data experts, field supervisors and tracer teams. The contact was defined as a person who interacted with a symptomatic MVD case (probable or confirmed) in at least one of the following manners: touched body fluids of the MVD case (blood, vomit, saliva, urine, feces, semen, tears, sweat), had direct physical contact with the body of the patient (alive or dead), touched or cleaned the linens, clothes, or dishes of the patient, or slept or ate in the same household with the patient. During the outbreak, people who met this definition were identified, isolated, and followed up. Identified contacts were documented using contact listing forms. Contacts who were healthcare workers were quarantined at designated hotels while other contacts were quarantined at home. Tracer team members visited contacts daily for 21 days assessing their status by asking about symptoms of MVD and contact information obtained was recorded in contact follow-up forms. Any contact who developed symptoms was re-classified as a suspect case and transferred to the Marburg Treatment Unit (MTU) for isolation and management. Contacts completing their 21-day follow-up were released from quarantine.

## Data collection

In this study data was extracted from November 13 to December 3, 2023 from information on MVD cases and contacts that were collected during the outbreak. Information was obtained from archived Case Investigation Forms, Contact Listing Forms, and Contact Follow-up forms. In addition, family members were contacted during data collection to gather information about the whereabouts of cases and any possible connections to other cases.

## Data management and analysis

Microsoft Excel (version 2019) was used to enter, clean, and analyze data. Epidemiological descriptive analysis based on time, place, and persons were performed. In addition, the QGIS software 3.26.3 was used to map the distribution of cases and contacts in the study area using openly available shape files accessed through Tanzania's National Bureau of Statistics. Results and findings are presented using narration, figures, tables, and maps.

## Ethical consideration

The information reported in this article represents data collected during the MVD outbreak response. An outbreak investigation is regarded as an emergency activity and was endorsed by

MOH officials and didn't get ethical approval however permission to analyze and publish this information was sought and granted from the Medical Research Coordinating Committee of the National Institute for Medical Research. Relatives of MVD cases provided oral consent before the interview. All personal information collected was treated with high confidentiality, and the presented data is anonymized.

## Results

### Timeline of onset of symptoms among cases

The probable index case started showing symptoms on February 27, 2023, and died on March 1, 2023. Six cases started showing symptoms between 9 to 14 days after the probable index started showing symptoms and four died. The final case displayed symptoms on April 11, 2023, no additional cases were reported for a continuous period of 42 days. Consequently, on June 1, 2023, the outbreak was officially declared to be over. The MVD epi curve with chronological date of onset and disease outcome is shown in Fig 1.

### Description and epidemiological linkage of MVD cases

A total of nine cases were reported of which eight were confirmed and one was probable. The age of cases ranged from 1–59 years with a median age of 34 years. Of all the cases, six were males, seven were family-related, and two were frontline healthcare workers. Six cases died, making the case fatality of 66.7%. The cases were identified based on the date seen at the health facility; onset of symptoms was based on the case investigation done during the response. Description for individual case reports is given below:

**Case 1:** A 33-year-old fisherman from Butayaibega village who conducts his fishing activities around the islands in Lake Victoria, where there is a significant population movement for fishing trade, including individuals from neighboring country Uganda. Moreover, the islands and Bukoba town where he lives have large caves that harbors bats, whose manure is used for small-scale farming.

On February 26, 2023, he returned from his fishing activities and attended a wedding in the same village. The next day, he started experiencing headache and fever. At first, his family believed these symptoms were related to previous health problems he had experienced, such as vomiting blood and abdominal pain, which had occurred about two weeks before this event.

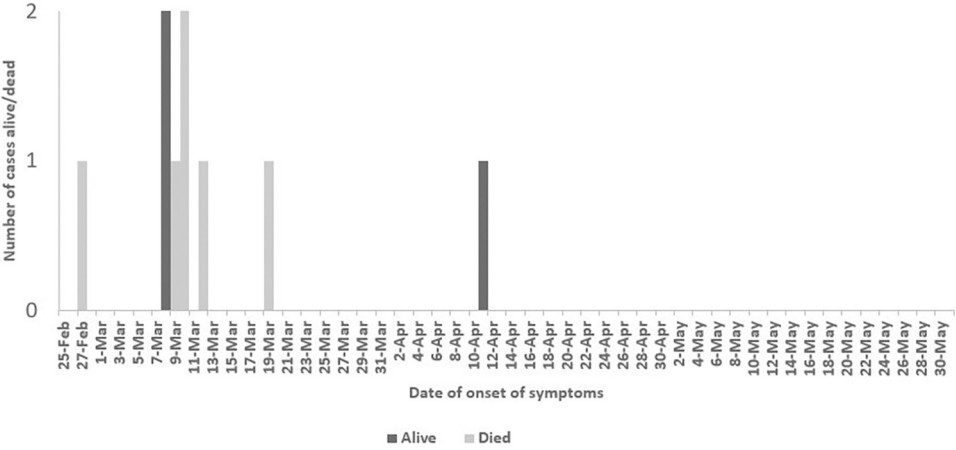

**Fig 1. Epi curve for MVD cases and deaths in Bukoba District, Kagera region, Tanzania from February 27 to May 31, 2023.**

Instead of seeking immediate medical care, the family opted for home care and moved the fisherman between relatives. He was cared for by his elderly uncle on February 27, 2023, and then transferred to his aunt's house on February 28, 2023, where his mother (Case 4), cousin (Case 6), and sister (Case 9) provided care. Two other sisters who were in close contact with him showed no symptoms during the 21-day follow-up period. By the night of February 28, 2023, the fisherman's condition had worsened significantly, with symptoms of vomiting blood and extreme weakness. He was then taken to the local Health Centre in Maruku with the help of villagers, where healthcare workers (Case 5 and Case 7) attended to him. A routine malaria rapid test was conducted by the laboratory personnel (Case 5). On March 1, 2023, he was referred to the Regional Referral Hospital (RRH) in Bukoba, accompanied by relatives, including case 3, in a private vehicle. He passed away on the way to RRH. No MVD laboratory sample was collected during his illness.

Following his death, his body was transported back to his uncle's house in the same vehicle used for his hospital transfer. Two relatives (added to the contacts) performed traditional cleaning rituals without any guidance or oversight from health care professionals trained in infection prevention and control (IPC); he was buried on March 2, 2023.

**Case 2:** This was a laboratory confirmed, 50-year-old female, who was the aunt of the probable index case (Case 1), lived in the same neighborhood, and cared for the probable index case at her home. She shared her household with 19 family members, including her son (Case 6). During the index case's illness, case 2 and her four daughters, residing in the same household, provided care.

Case 2 was initially in good health. Starting on March 10, she developed symptoms including headaches, fatigue, and vomiting. She sought medical assistance and was temporarily admitted to the local Health Centre for observation on March 12, 2023. She was later discharged with medication to be taken at home. On March 13, 2023, her sister-in-law (Case 4) visited her to check on her well-being, and she mentioned that she was also feeling unwell. Despite being discharged with treatment instructions, her condition worsened on March 14, 2023 and she was readmitted to the local Health Centre before being transferred to RRH in Bukoba.

Case 2 symptoms at RRH included fever, general body weakness, and vomiting blood, with a diagnosis of upper gastrointestinal bleeding. She passed away on March 15, 2023. Nasopharyngeal swab for laboratory testing was collected on March 16, 2023 while deceased and the results were available on the same day. Her body was temporarily kept at the RRH mortuary and received a supervised dignified burial on March 16, 2023.

**Case 3:** The second laboratory confirmed case was a 27-year-old man who was the cousin of the probable index case. He lived in Maruku and operated a local small business. He escorted the probable index case while being transferred from first local Health Centre to RRH in Bukoba. His illness began on March 9, 2023, with unspecified initial symptoms. He sought medical attention at first local Health Centre on March 12, 2023, and was referred to local Town Health Centre. From there, he was eventually transferred to RRH. By then he was presenting with bleeding from various orifices and episodes of convulsion.

He passed away on March 16, 2023, and his body was temporarily kept at the RRH mortuary. Nasopharyngeal swab for laboratory testing was collected on March 16, 2023, while deceased and the results were available on the same day. He received a dignified and supervised burial on March 17, 2023.

**Case 4:** The third laboratory confirmed case was a 59-year-old female and the mother of the probable index case. She lived in the same neighborhood as Case 2, where the probable index case received care while ill. She started experiencing symptoms of fever, headache, vomiting, abdominal pain, lethargy, anorexia, and difficulty breathing on March 12, 2023.

On March 13, 2023, she sought medical assistance at a first local dispensary, using a private motorcycle ("Boda-boda") ride as transportation. Her daughter (Case 9) accompanied her during this visit. However, her condition worsened on March 14, 2023, and she returned to the first local dispensary, again using the same Boda-boda. She was then referred to Town Health Centre. Unfortunately, she passed away on March 15, 2023. Nasopharyngeal swab for laboratory testing was collected on March 16, 2023 while deceased and the results were available on the same day. She received a supervised and dignified burial on March 16, 2023.

**Case 5:** The fourth laboratory confirmed case was a 47-year-old male health worker who conducted routine laboratory testing (i.e., malaria rapid test) for the probable index case on March 1, 2023. On March 10, 2023, he began to experience symptoms of illness, including fatigue, fever, headache, and vomiting blood. March 13, 2023, he sought medical attention and was admitted to the Town Health Centre. As his condition worsened, he was transferred to the RRH on March 15, 2023, where he received care from multiple healthcare workers equipped with standard personal protective equipment (PPE). Unfortunately, he passed away on March 16, 2023. Nasopharyngeal swab for laboratory testing was collected on March 16, 2023 while deceased and the results were available on the same day., He received a supervised and dignified burial.

**Case 6:** The fifth laboratory confirmed case was a 26-year-old male, who is the son of Case 2 and a cousin to the probable index case. He lived in the same household as his mother (Case 2), where the probable index case received care while ill. On March 8, 2023, he developed symptoms, including fever, headache, and general body weakness. In response to his symptoms, he was initially admitted to second local Dispensary (in Ntoma) on March 9, 2023, for one day but was discharged on March 10, 2023, with instructions to continue treatment as an outpatient. However, on March 15, 2023, his condition worsened and was readmitted to the first local Health Centre. On March 16, 2023, he was confirmed to be MVD case and was transferred to the Marburg Treatment Unit in Kabyaile and continued with treatment. He responded well to treatment and survived; he was discharged on April 4, 2023.

**Case 7:** The sixth laboratory confirmed case was a 35-year-old male, who was a health care worker, and provided care to probable index case, case 4 and case 5 from February 28,2023 onwards. He began showing symptoms on March 8, 2023, including fever, headache, diarrhea, and vomiting blood. Despite feeling unwell, he continued to provide care to other patients.

At first, after feeling ill, he visited the nearby health facility, where he was diagnosed and treated for malaria with Artemether Lumefantrine (ALU). Without improvement, his treatment was changed to high-dose Gentamycin injections.

On March 15, he was referred to the RRH where acute renal failure was diagnosed. On March 16, a sample was taken, and he was confirmed to have MVD. He was isolated and received treatment at the RRH, eventually made a full recovery, on May 22,2023 he was discharged to go home.

**Case 8:** This seventh laboratory confirmed case was an 18-month-old male child, who was the grandson of Case 4 and the son of Case 9. He was being monitored as a contact of Case 4, who was deceased. On March 19, 2023, the child began to exhibit fever and loss of appetite.

Following the development of symptoms, he tested positive for MVD on March 21, 2023 and was admitted to the MTU on the same day. Despite the medical care, he passed away on April 10, 2023, and received a supervised and dignified burial.

**Case 9:** The last and eighth laboratory confirmed case was a 38-year-old female, who is the mother of Case 8. She was one of the 212 contacts as she took care of both the probable index case (her brother) and Case 4 (her mother). Due to her child's young age (Case 8), she had to be admitted with her child to continue caring for him, which is likely how she was exposed to the virus.

On April 10, 2023, she presented with low grade fever, headache, body fatigue, nausea, loss of appetite, vomiting, and heavy vaginal bleeding mixed with clotted blood. She tested positive for MVD on the same day. Additionally, received psychological and social support. Eventually, she recovered and on April 19, 2023, she was discharged as a survivor of MVD.

## Description of cases by place

The outbreak was confined to Kagera region in Bukoba District specifically in Maruku and Kanyangereko wards. Moreover, among seven villages in two wards, 5 (55.6%) cases were from Butayaibega village located in Kanyangereko ward (Fig 2).

## Epidemiological description of contacts

A total of 212 individuals were identified as MVD contacts. Females (n = 115 (54.2%)) contributed to the majority of contacts. Contacts between 15–29 and 30–44 years of age accounted for 35.8% and 35.4% of contacts, respectively. Regarding the risk of exposure, 116 (37.9%) and 67 (21.9%) contacts had direct physical contact and body fluid exposure from cases respectively.

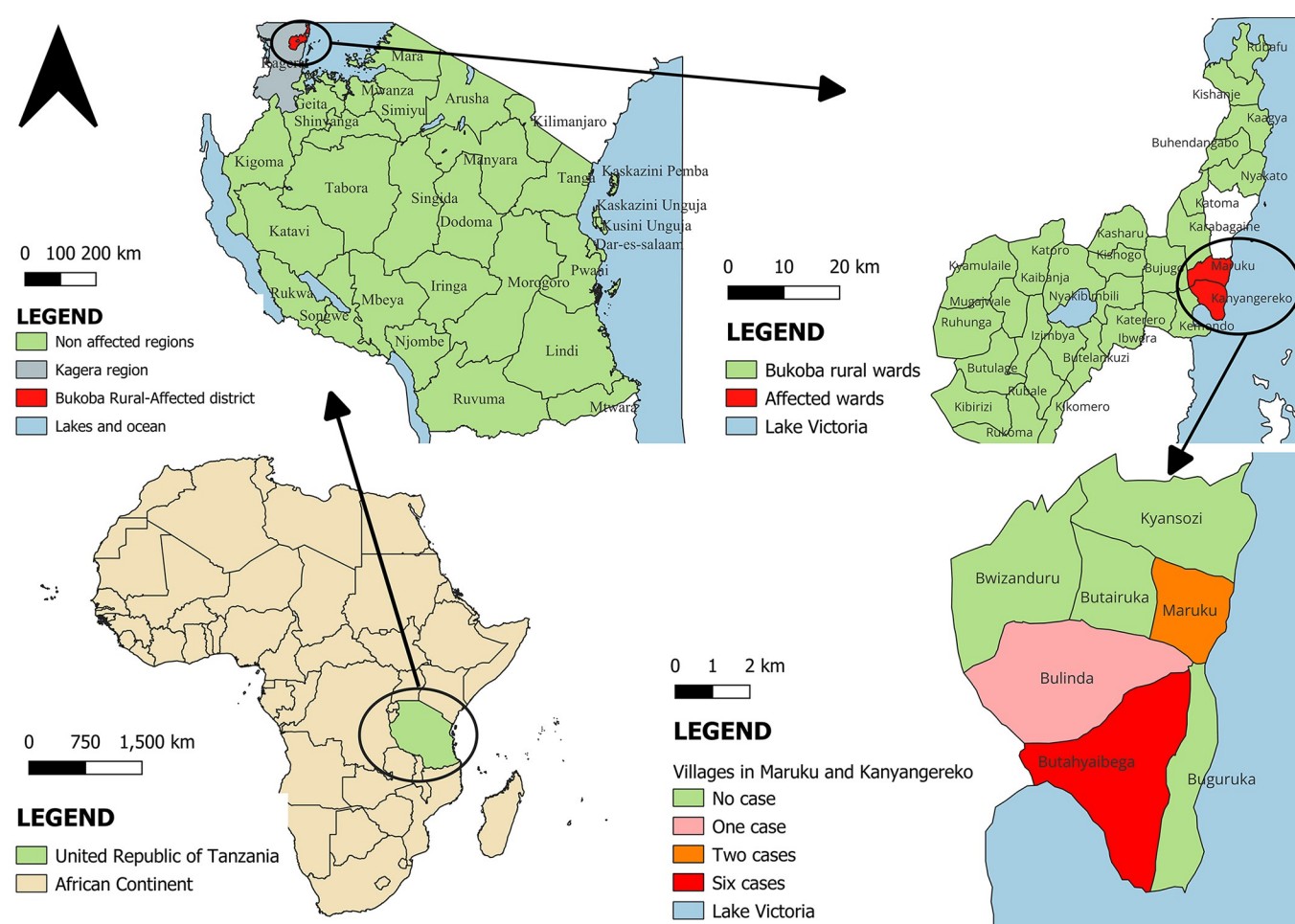

**Fig 2. A map showing distribution of MVD cases in Kagera region, Tanzania from February 27 to May 31, 2023.** Map was drawn using QGIS desktop software 3.26.3. The shapefiles used were from an openly available source (https://www.nbs.go.tz/statistics/topic/gis). The shapefiles were made based on the 2012 population and housing census, but in this study, shapefiles have been modified to capture all the regions information.

**Table 1. Characteristics of contacts in Bukoba District, Kagera region from March to May 2023 (n = 212).**

| Variable | Frequency (Percentage) |
|---|---|
| **Sex** | |
| Female | 115 (54.2) |
| Male | 97 (45.8) |
| **Age group (years)** | |
| ≤5 | 14 (6.6) |
| 6–14 | 12 (5.7) |
| 15–29 | 76 (35.8) |
| 30–44 | 75 (35.4) |
| 45–59 | 27 (12.7) |
| ≥60 | 8 (3.8) |
| **Type of exposure* (n = 306) ** ** | |
| 1 | 67 (21.9) |
| 2 | 116 (37.9) |
| 3 | 39 (12.7) |
| 4 | 84 (27.5) |
| **Relation of contact to case** | |
| Family members | 87 (41.0) |
| Friends | 10 (4.7) |
| HCWs | 89 (42.0) |
| Patients in the same wards | 23 (10.8) |
| Relatives in the ward | 2 (0.9) |
| Security guard | 1 (0.5) |
| **Outcome** | |
| Graduated | 209 (98.6) |
| Died due to other cause | 1 (0.5) |
| Contact who become case | 2 (0.9) |
| **Districts** | |
| Bukoba District Council | 117 (55.2) |
| Bukoba Municipal Council | 95 (44.8) |

* Types of Exposure

1 = Touched body fluids of the MVD case (blood, vomit, saliva, urine, feces, semen, sweat)

2 = Had direct physical contact with the body of the patient (alive or dead)

3 = Touched or cleaned the linens, clothes, or dishes of the patient

4 = Slept or ate in the same household as the patient

** Other contacts had more than one exposure type

Most of the contacts were HCWs (n = 89 (42.05%)) followed by family members (n = 87 (41.0%)). Over the course of the follow-up period, 2 (0.9%) developed symptoms and tested positive for MVD. The majority of contacts (n = 117 (55.2%)) were from Bukoba District Council, and most of them resided in Kanyangereko ward (n = 64 (30.3%)) (Table 1 & Fig 3).

## Discussion

This study describes the first MVD outbreak in Tanzania involving a cluster of cases among closely related family members and frontline healthcare workers. Cases did not spread beyond the initial cluster, a scenario similar to one observed in Eastern Uganda that affected four close family members [12]. Seventeen days passed between the possible onset of illness in the

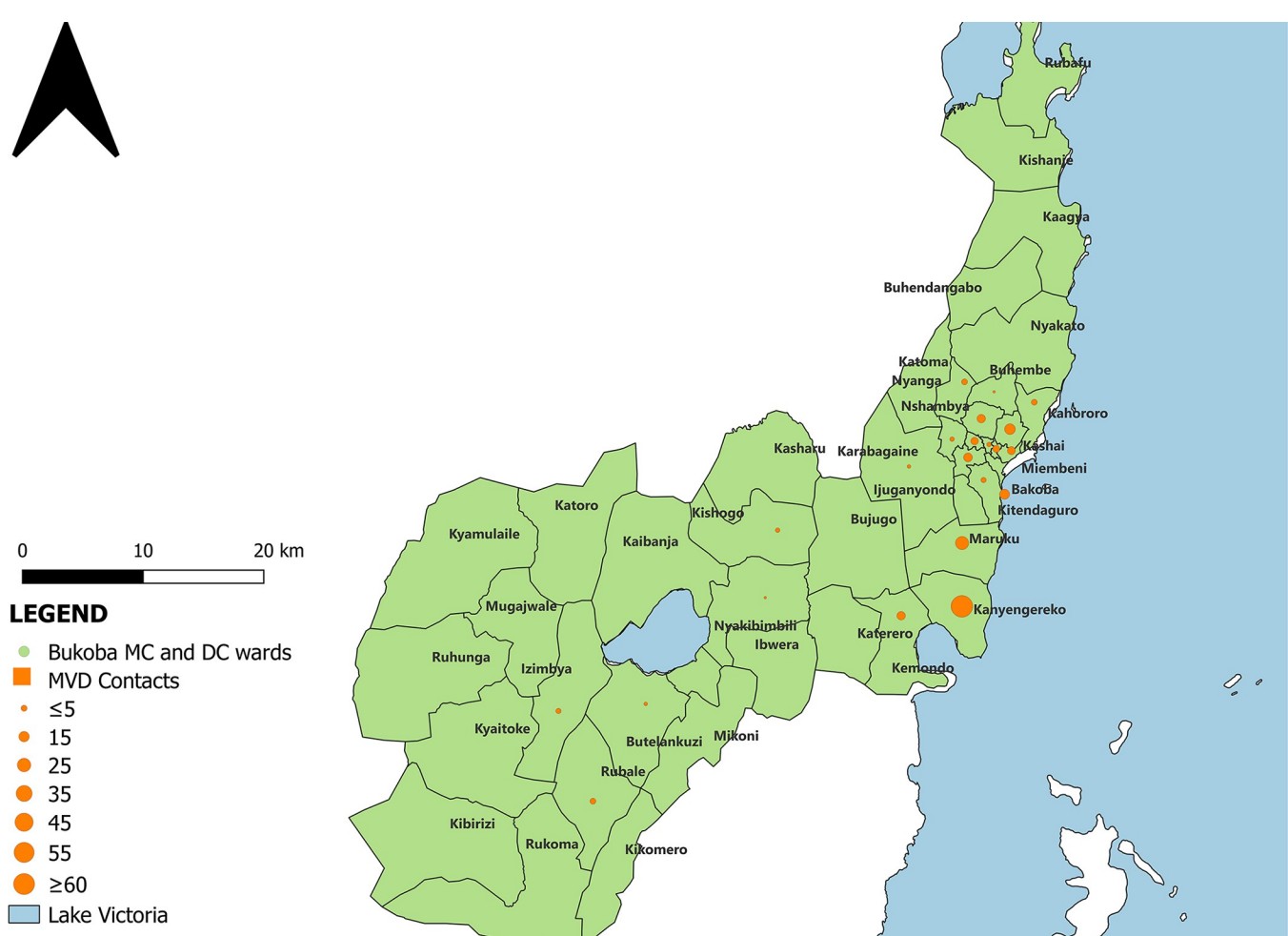

**Fig 3. Distribution of MVD contacts in Bukoba Districts, Kagera region, Tanzania from February 27 to May 31, 2023.** Map was drawn using QGIS desktop software 3.26.3. The shapefiles used were from an openly available source (https://www.nbs.go.tz/statistics/topic/gis). The shapefiles were made based on the 2012 population and housing census, but in this study, shapefiles have been modified to capture all the regions information.

probable index case until laboratory confirmation of subsequent cases. Immediate families were most vulnerable with at least four succumbing to the disease within a span of three days, a scenario that greatly limited potential investigation of exact movements and exposure risks of the initial probable case. Possible initial zoonotic transmission therefore remains unknown, although fruit bats in caves around Bukoba and in other Islands in Lake Victoria where the probable index case may have been fishing, are a common sight. Reports from western Uganda, an area close to Bukoba with reported MVD outbreak in 2015, also document the presence of Rousettus bats that harbor Marburg virus [22].

This outbreak reported a case fatality rate of 67% which is relatively higher when compared to the rates reported in Germany in 1967 (22.6%), Kenya in 1980 (33%), and Uganda in 2012 (58%) [7, 14, 23]. However, higher case fatality rates have been reported in Uganda in 2017 (75%), Angola in 2005 (88%), and DRC Congo in 1998 (83%) [11, 13, 16]. Several factors might have contributed to the observed high case fatality rate. The outbreak was confirmed 17 days after the probable index case showed symptoms despite the probable index and subsequent cases seeking treatment at several healthcare facilities on March 1 and March 8, 2023. Our detailed findings show the low index of suspicion among healthcare workers in all the

health facilities that managed the initial cases. One possible explanation for this delay could be due to initial symptoms of MVD often mimicking those of other common endemic diseases prevalent in our local area, such as malaria and typhoid fever [24]. As a result, misdiagnosis can occur and lead to a delay in recognizing MVD as the root cause of the illness.

During this outbreak, only two individuals who had close contact with family members who died from the disease developed symptoms and became cases. This could be attributed to the low transmission rate of the disease in its early stages or was undetected during the outbreak as the Marburg virus can persist in the semen and vaginal excretions of survivors [25]. On the other hand, prompt response within three days following confirming of the outbreak could have played a role in halting the transmission through close monitoring of all identified contacts.

Since the Rift Valley Fever outbreak in 2008, this is the first confirmed VHF outbreak in recent years in Tanzania. The country was able to utilize health security capacities, such as event-based surveillance (EBS), to identify the unknown disease cluster and quickly confirm, respond, and control the MVD by leveraging multisectoral emergency preparedness and response systems. Capacities in place included presence of a mobile laboratory, treatment and isolation facilities, surge training of HCWs on emergency preparedness and response, and trained community health workers on EBS and transportation resources.

Furthermore, the long-standing health systems strengthening from HIV/Tuberculosis, Malaria, and vaccine preventable diseases established service delivery infrastructure. These resources were leveraged to enhance surveillance, scale up infection prevention and control and support case management for better patient outcomes.

Despite the health security capacities, the delay in detection underscores the need for robust and sustained early warning alert and response systems. It is possible the operational VHF readiness efforts in Kagera were waning following the declaration of the end of SVD outbreak in Uganda on January 11, 2023. To strengthen early detection, it is crucial to improve awareness at health facilities about the symptoms of VHF and the associated lab testing to confirm a diagnosis. In addition, the adoption of optimal metrics such as the 7-1-7 approach which aims to measure the detection within 7 days of occurrence, notification of relevant authorities within 1 day and implementation of response measures within 7 days [26] can help identify bottlenecks in response efforts. Moreover, investigations on potential reservoirs and zoonotic transmission dynamics are needed to strengthen health security infrastructure and improve preparedness, readiness, and response in the future.

The inability to identify the origin of the MVD can pose a substantial challenge in terms of preventing future outbreaks.

## Conclusion

The transmission of this outbreak was limited to family members and healthcare workers who provided care to the cases. The delay in detection aggravated the spread and possibly the consequent fatality but once confirmed the swift response stemmed further transmission. The outbreak lasted for 72 days and was contained within two neighboring wards in Bukoba District. There is a need to strengthen early detection and reporting at all levels. Further research to investigate the ecological epidemiology of the Marburg virus within Kagera region is highly recommended.

## Supporting information

**S1 Data.**
(XLSX)

## Acknowledgments

We acknowledge Ministry of Health and President Office Regional Administration Local Government for their swift efforts in containing the Marburg outbreak. Special thanks go to our development partners who actively supported the Tanzanian Ministry of Health during this response. We commend the Kagera Regional/Council Health Management Team and the Mobile Laboratory Staff in Kagera for their leadership and vital contributions in initial investigation of the outbreak, ensuring its containment. We also acknowledge and appreciate the dedication of field supervisors and tracer teams in contact tracing. Lastly, we extend our thanks to the affected villages' community members and their families for their active role in controlling the outbreak.

**Disclaimer:** The findings and conclusions in this paper are those of the authors and do not necessarily represent the views of the U.S. Centres for Disease Control and Prevention.

## Author Contributions

**Conceptualization:** Vida Mmbaga, George Mrema, Danstan Ngenzi, Emmanuel Mwakapasa, Hamza Matimba, Gerald Manase, Christer Kanyankole, Ishata Conteh.

**Data curation:** Vida Mmbaga, George Mrema, Danstan Ngenzi, Welema Magoge, Emmanuel Mwakapasa, Frank Jacob, Hamza Matimba, Medard Beyanga, Angela Samweli, Michael Kiremeji, Mary Kitambi, Erasto Sylvanus, Ernest Kyungu, Joseph Hokororo, Christer Kanyankole, Martin Rwabilimbo, George Kauki, Maria Ezekiely Kelly, William Mwengee, Faraja Msemwa, Kokuhabwa Mukurasi, Marcelina Mponela.

**Formal analysis:** Vida Mmbaga, George Mrema, Danstan Ngenzi, Welema Magoge, Emmanuel Mwakapasa, Angela Samweli, Eliakimu Kapyolo.

**Investigation:** Vida Mmbaga, George Mrema, Danstan Ngenzi, Welema Magoge, Emmanuel Mwakapasa, Frank Jacob, Hamza Matimba, Medard Beyanga, Angela Samweli, Michael Kiremeji, Mary Kitambi, Erasto Sylvanus, Ernest Kyungu, Joseph Hokororo, Martin Rwabilimbo, Issessanda Kaniki, George Kauki, Maria Ezekiely Kelly, William Mwengee, Gabriel Ayeni, Faraja Msemwa, George S. Mgomella, Kokuhabwa Mukurasi, Marcelina Mponela, Wangeci Gatei.

**Methodology:** Vida Mmbaga, George Mrema, Danstan Ngenzi, Welema Magoge, Emmanuel Mwakapasa, Frank Jacob, George Kauki, Marcelina Mponela, Eliakimu Kapyolo, Jonathan Mcharo, Mary Mayige, Pius Horumpende.

**Supervision:** Vida Mmbaga, Medard Beyanga, Angela Samweli, Michael Kiremeji, Mary Kitambi, Erasto Sylvanus, Gerald Manase, Grace Saguti, George S. Mgomella, Wangeci Gatei, Ishata Conteh, Peter Mala, Mahesh Swaminathan, Paschal Ruggajo, Grace Magembe, Zabulon Yoti, Elias Kwesi, Tumaini Nagu.

**Validation:** Vida Mmbaga, Medard Beyanga, Angela Samweli, Michael Kiremeji, Mary Kitambi, Erasto Sylvanus, Gerald Manase, Issessanda Kaniki, Gabriel Ayeni, Grace Saguti, Mary Mayige, Wangeci Gatei, Ishata Conteh, Peter Mala, Mahesh Swaminathan, Pius Horumpende, Paschal Ruggajo, Grace Magembe, Zabulon Yoti, Elias Kwesi, Tumaini Nagu.

**Visualization:** Vida Mmbaga, George Mrema, Danstan Ngenzi, Welema Magoge, Emmanuel Mwakapasa, Frank Jacob, Hamza Matimba, Medard Beyanga, Angela Samweli, Michael Kiremeji, Mary Kitambi, Erasto Sylvanus, Ernest Kyungu, Gerald Manase, Joseph Hokororo, Christer Kanyankole, Martin Rwabilimbo, Issessanda Kaniki, George Kauki, Maria Ezekiely Kelly, William Mwengee, Gabriel Ayeni, Faraja Msemwa, Grace Saguti, George S.

Mgomella, Kokuhabwa Mukurasi, Marcelina Mponela, Eliakimu Kapyolo, Jonathan Mcharo, Mary Mayige, Wangeci Gatei, Ishata Conteh, Peter Mala, Mahesh Swaminathan, Pius Horumpende, Paschal Ruggajo, Grace Magembe, Zabulon Yoti, Elias Kwesi, Tumaini Nagu.

**Writing – original draft:** Vida Mmbaga, George Mrema, Danstan Ngenzi, Welema Magoge, Emmanuel Mwakapasa, Hamza Matimba, Gerald Manase, Christer Kanyankole.

**Writing – review & editing:** Vida Mmbaga, George Mrema, Danstan Ngenzi, Welema Magoge, Emmanuel Mwakapasa, Frank Jacob, Hamza Matimba, Medard Beyanga, Angela Samweli, Michael Kiremeji, Mary Kitambi, Erasto Sylvanus, Ernest Kyungu, Gerald Manase, Joseph Hokororo, Christer Kanyankole, Martin Rwabilimbo, Issessanda Kaniki, George Kauki, Maria Ezekiely Kelly, William Mwengee, Gabriel Ayeni, Faraja Msemwa, Grace Saguti, George S. Mgomella, Kokuhabwa Mukurasi, Marcelina Mponela, Eliakimu Kapyolo, Jonathan Mcharo, Mary Mayige, Wangeci Gatei, Ishata Conteh, Peter Mala, Mahesh Swaminathan, Pius Horumpende, Paschal Ruggajo, Grace Magembe, Zabulon Yoti, Elias Kwesi, Tumaini Nagu.

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
