## [Decision Letter · Decision Letter 0]

15 Apr 2024

PONE-D-24-07742Epidemiological description of Marburg virus disease outbreak in Kagera region, Northwestern TanzaniaPLOS ONE

Dear Dr. Mrema,

Thank you for submitting your manuscript to PLOS ONE. After careful consideration, we feel that it has merit but does not fully meet PLOS ONE’s publication criteria as it currently stands. Therefore, we invite you to submit a revised version of the manuscript that addresses the points raised during the review process.

 Please submit your revised manuscript by May 30 2024 11:59PM. If you will need more time than this to complete your revisions, please reply to this message or contact the journal office at plosone@plos.org. Please include the following items when submitting your revised manuscript:A rebuttal letter that responds to each point raised by the academic editor and reviewer(s). You should upload this letter as a separate file labeled 'Response to Reviewers'.A marked-up copy of your manuscript that highlights changes made to the original version. You should upload this as a separate file labeled 'Revised Manuscript with Track Changes'.An unmarked version of your revised paper without tracked changes. You should upload this as a separate file labeled 'Manuscript'.If applicable, we recommend that you deposit your laboratory protocols in protocols.io to enhance the reproducibility of your results. Protocols.io assigns your protocol its own identifier (DOI) so that it can be cited independently in the future. For instructions see: https://journals.plos.org/plosone/s/submission-guidelines#loc-laboratory-protocols. Additionally, PLOS ONE offers an option for publishing peer-reviewed Lab Protocol articles, which describe protocols hosted on protocols.io. Read more information on sharing protocols at https://plos.org/protocols?utm_medium=editorial-email&utm_source=authorletters&utm_campaign=protocols.

We look forward to receiving your revised manuscript.

Kind regards,

Masahiro Kajihara, PhD

Academic Editor

PLOS ONE

Journal Requirements:

3. For studies involving third-party data, we encourage authors to share any data specific to their analyses that they can legally distribute. PLOS recognizes, however, that authors may be using third-party data they do not have the rights to share. When third-party data cannot be publicly shared, authors must provide all information necessary for interested researchers to apply to gain access to the data. (https://journals.plos.org/plosone/s/data-availability#loc-acceptable-data-access-restrictions) 

4. We note that Figures 3 and 4 in your submission contain map/satellite images which may be copyrighted. All PLOS content is published under the Creative Commons Attribution License (CC BY 4.0), which means that the manuscript, images, and Supporting Information files will be freely available online, and any third party is permitted to access, download, copy, distribute, and use these materials in any way, even commercially, with proper attribution. For these reasons, we cannot publish previously copyrighted maps or satellite images created using proprietary data, such as Google software (Google Maps, Street View, and Earth). For more information, see our copyright guidelines: http://journals.plos.org/plosone/s/licenses-and-copyright.

a. You may seek permission from the original copyright holder of Figures 3 and 4 to publish the content specifically under the CC BY 4.0 license.  

Additional Editor Comments:

As highlighted by the reviewers, the manuscript contains a considerable number of editorial and typographical errors, including incomplete sentences and grammatical inaccuracies. Given the significant impact of these issues on the clarity and professionalism of the manuscript, I strongly recommend securing editorial assistance from a proficient English editor well-versed in manuscript formatting best practices. This thorough proofreading process will greatly enhance the quality of the manuscript for the revision process.

Additionally, I kindly request addressing the following minor points, along with considering the reviewers' comments:

1. Please verify the number of samples listed under "Type of exposure" in Table 1. Is it indeed 309, or should it be 306?

2. Ensure that scales are provided for each map in the figures.

3. In Figure 1, please provide an explanation of the numerical values depicted.

Thank you for your attention to these matters.

Reviewers' comments:

Reviewer's Responses to Questions

**Comments to the Author**

1. Is the manuscript technically sound, and do the data support the conclusions?

Reviewer #1: Yes

Reviewer #2: Yes

2. Has the statistical analysis been performed appropriately and rigorously? 

Reviewer #1: N/A

Reviewer #2: N/A

3. Have the authors made all data underlying the findings in their manuscript fully available?

Reviewer #1: No

Reviewer #2: No

4. Is the manuscript presented in an intelligible fashion and written in standard English?

Reviewer #1: No

Reviewer #2: Yes

5. Review Comments to the Author

Reviewer #1: Epidemiological description of Marburg virus 1 disease outbreak in Kagera region, Northwestern Tanzania

The authors present an epidemiological description of the MVD outbreak in Tanzania. This is an interesting account of the outbreak response, the cases and circumstances surrounding each case. The manuscript would do with a few minor changes as well editing as explained below.

4. IS THE MANUSCRIPT PRESENTED IN AN INTELLIGIBLE FASHION AND WRITTEN IN STANDARD ENGLISH?

No, it would benefit from some editing of grammatical and typographical errors, as well as rewriting/rephrasing some sentences. Some are noted below.

Page 3:

(i) sentence beginning line 63 to 64 - The etiologies of MVD are RNA viruses………. is incomplete

(ii) sentence beginning line 74 to 77 - The SUDV recorded 164 cases…….. should be rephrased, e.g. There were 164 cases of EVD (142 confirmed and 22 probable) and….

(iii) line 77: Kagera Region - region is capitalised, but through out the manuscript it is all lowercase

Page 5:

(i) line 158: The last sentence is incomplete

(ii) line 161: “information about on…”

Page 7:

(i) line 213: personal (case 5)

Page 8:

(i) sentence beginning line 227: On March 13, 2023, her sister-in-law (Case 4), who was the mother of the index case, visited her, but she also reported feeling unwell.

Page 11:

(i) line 310: here and in other parts of the manuscript fig should be capitalized. Also, if it is in text, it should not be bold.

(ii) Line 322: either should be used with or. In this case either can be removed

(iii) Line 325: “And the majority”

Page 14:

(i) Line 394: there are 2 full stops

(ii) Line 404: the sentence “To strengthen early detection and reporting at all levels.” Seems to be missing something at the beginning.

OTHER COMMENTS

There have been changes to filovirus taxonomy. The authors should follow the International Committee for Taxonomy of Viruses instructions for the taxonomy of Filoviruses. (Biedenkopf, N.; Bukreyev, A.; Chandran, K.; Di Paola, N.; Formenty, P. B. H.; Griffiths, A.; Hume, A. J.; Mühlberger, E.; Netesov, S. V.; Palacios, G.; Pawęska, J. T.; Smither, S.; Takada, A.; Wahl, V.; Kuhn, J. H., Renaming of genera Ebolavirus and Marburgvirus to Orthoebolavirus and Orthomarburgvirus, respectively, and introduction of binomial species names within family Filoviridae. Arch Virol 2023, 168, (8), 220. https://doi.org/10.1007/s00705-023-05834-2.)

Page 3:

Line 73: I suggest it read: “On September 20th, 2022, an outbreak of Ebola virus disease (EVD) caused by Sudan Virus (SUDV) was declared by the government of Uganda in Mubende district Western Uganda.”

This will also allow for the definition of EVD that is in line 83, as in the manuscript it is introduced for the first time, but is not written out in full.

Line 77: I think it is important to indicate that Mutukula town is in Tanzania. This means that line 78 can end “90 km (55 miles) from Mutukula town.” And delete “on the Kagera region (Tanzania), border.”

Line 83: EVD is introduced for the first time, but is not written out in full. If

I think it would be better to add a sentence indicating that EVD and MVD have similar presentation.

Page 7:

Line 188: refers the index. I presume this should read the probable index case. Here and through out the manuscript, it would be better to refer to this case as probable index case and not index case.

Line 204: this is interesting past medical history of vomiting blood. Was there any indication on the timing? Was it the recent past? It may be better to indicate.

Page 10:

Case 7: I think it would be better to indicate form the 1st sentence that Case & treated the index case. This gives a clearer epidemiological picture. Therefore “Index case on 28th February” should move from line 280 to a more appropriate line.

Page 13:

Line 349: it should be clear that the laboratory confirmation was not from the probable index case

Lines 357-360: in this section, the outbreak year has only been given for Uganda. Is there a specific reason? If not, it is better to be consistent and indicate the outbreak year for all or none.

Lines 363-367: it has been documented in literature that ignorance of health care workers about the disease is one of the reasons there is a delay in detection of an outbreak of EVD/MVD, particularly in areas where no previous outbreak has occurred. It would be good to make reference to this.

Lines 368-373: the basic reproduction number for Marburg virus is relatively low. It would be good to make a comment on this aspect.

Line 374-375: best to put (VHF) after viral hemorrhagic fever as it is the first time it is appearing in the manuscript.

Page 14:

Line 388: It would be better to replace SUDV with “the EVD outbreak”

Reviewer #2: line70: the marburg outbreak in Equatorial Guinea occured in 2023...not 2021

line 37-38. What does the sentence "laboratory testing used altona diagnostics realstart filovirus screen RTPCR with at least 2 tests in two separate laboratories" mean? do you mean that the sample sample had to be tested using the same assay in two geographically separate laboratories? if so, why?

line: 156: I think there might be a typo....currently it says that "any contacts who developed symptoms were re-classified as a case....i think you mean "suspect"

line 158: the last sentence is incomplete

line 163: did the reach out to close family members of the individuals involved occur during the outbreak? or afterwards?

Description and epidemiologic linkage of MVD cases: it might be best to summarize the key findings for each patient and place the detailed summaries in the appendix? Also, it would be helpful to provide additional epi details on case 1 (such as his occupation and details on how he might have been infected...some of this information is currently in the discussion section)

line 322-323: what is the difference betweeen family members and relatives?

line 359: would suggest not including the CFR for the Netherlands as it was just 1 patient.

Fig 2: not sure if this figure is necessary

6. PLOS authors have the option to publish the peer review history of their article (what does this mean?). If published, this will include your full peer review and any attached files.

Reviewer #1: No

Reviewer #2: No

---

## [Author Response · Author response to Decision Letter 0]

28 May 2024

-We would like to thank the editor and reviewers for their helpful feedback on our manuscript, "Epidemiological description of Marburg virus disease outbreak in Kagera region, Northwestern Tanzania". Their comments have significantly improved the quality of our work, and we have carefully considered each suggestion in the revised version of the paper. We are confident that we have adequately addressed the raised concerns and that the manuscript is now ready for submission

---

## [Decision Letter · Decision Letter 1]

19 Jun 2024

PONE-D-24-07742R1Epidemiological description of Marburg virus disease outbreak in Kagera region, Northwestern TanzaniaPLOS ONE

Dear Dr. Mrema,

Thank you for submitting your manuscript to PLOS ONE. After careful consideration, we feel that it has merit but does not fully meet PLOS ONE’s publication criteria as it currently stands. Therefore, we invite you to submit a revised version of the manuscript that addresses the points raised during the review process.

We look forward to receiving your revised manuscript.

Kind regards,

Masahiro Kajihara, PhD

Academic Editor

PLOS ONE

Journal Requirements:

Additional Editor Comments:

Regarding my comment 2, "Ensure that scales are provided for each map in the figures," I can accept the current figures even though map scales are not provided (of course, it is better to include them since not all readers are familiar with Tanzania's geology). However, the shapefiles are not available from the URL provided in the legend. Please check the URL again.

Reviewers' comments:

Reviewer's Responses to Questions

**Comments to the Author**

1. If the authors have adequately addressed your comments raised in a previous round of review and you feel that this manuscript is now acceptable for publication, you may indicate that here to bypass the “Comments to the Author” section, enter your conflict of interest statement in the “Confidential to Editor” section, and submit your "Accept" recommendation.

Reviewer #1: (No Response)

Reviewer #2: All comments have been addressed

2. Is the manuscript technically sound, and do the data support the conclusions?

Reviewer #1: Yes

Reviewer #2: Yes

3. Has the statistical analysis been performed appropriately and rigorously? 

Reviewer #1: N/A

Reviewer #2: N/A

4. Have the authors made all data underlying the findings in their manuscript fully available?

Reviewer #1: Yes

Reviewer #2: Yes

5. Is the manuscript presented in an intelligible fashion and written in standard English?

Reviewer #1: No

Reviewer #2: Yes

6. Review Comments to the Author

Reviewer #1: My recommendations on the nomenclature of Filoviruses have not been addressed adequately. The lines 61 to 64 need to be rewritten to convey the correct nomenclature.

The authors should follow the International Committee for Taxonomy of Viruses instructions for the taxonomy of Filoviruses. (Biedenkopf, N.; Bukreyev, A.; Chandran, K.; Di Paola, N.; Formenty, P. B. H.; Griffiths, A.; Hume, A. J.; Mühlberger, E.; Netesov, S. V.; Palacios, G.; Pawęska, J. T.; Smither, S.; Takada, A.; Wahl, V.; Kuhn, J. H., Renaming of genera Ebolavirus and Marburgvirus to Orthoebolavirus and Orthomarburgvirus, respectively, and introduction of binomial species names within family Filoviridae. Arch Virol 2023, 168, (8), 220. https://doi.org/10.1007/s00705-023-05834-2.)

Line 61:

Marburg virus disease has not been renamed as Orthomarburgvirus. Marburg virus disease is caused by the orthomarburgviruses Marburg virus and Ravn virus.

Line 62/63:

The sentence: ‘The virus belongs to the Filoviridae family, which is the same family as the Ebola virus’ is wrong. Ebola virus is one of the orthoebolaviruses and is the representative virus of the species Orthoebolavirus zairense. This sentence needs correct nomenclature as well as English language editing.

Line 63/64:

The sentence ‘Fruit bats (Rousettus aegyptiacus) are widely believed to be the reservoirs of Marburg viruses’

It has been established that R. aegyptiacus are reservoirs of orthomarburgviruses, so it is incorrect to state that they are widely believed to be reservoirs.

Line 77/78:

‘Since EVD and MVD belong to the same family, they share common symptoms and are both transmitted through contact.’

This sentence implies that all viruses belonging to the family Filoviridae share common symptoms. But this is not the case. It would be better to rephrase the sentence to just state that EVD and MVD have common symptoms. Also add a citation for this.

Line 161 -will benefit from English language editing

Line 390: the term SUDV was not replaced by EVD. What is in the manuscript is ‘Ebola’ and not EVD

Reviewer #2: line 36 (abstract): spell out RT-PCR

line 37: instead of "physically followed up" consider followed up in-person

line 43: consider also providing median age as there are extremes in the age range

line 62: consider deleting "and non-human primates" as this list is not exhaustive

line 79: The outbreak in Uganda was due to Sudan virus, so use Sudan virus disease instead of EVD. EVD now only refers to disease due to Ebola virus (species Orthoebolavirus zairense)

line 82: viral hemorrhagic fever should not be capitalized

line 120: for clarity, consider cutting "having any of the following"

line 137: considering using the term orthoebolaviruses instead of EVD as the Altona can detect multiple different orthoebolaviruses

line 137: If you decide to use orthoebolaviruses instead of EVD, considering using the term marburg virus instead of MVD so that you are listing viruses

line 197: consider providing median age (as comment above)

line 225: clarify what you mean by "without any expert supervision" ...i assume this means without IPC supervision?

line 369: malaria and typhoid fever should not be capitalized

line 371: please note that pauci-symptomatic marburg virus disease has been detected in Uganda during follow up serosurveys. Thus it is possible there were more people infected, but were not detected during the outbreak. This is especially important given that Marburg virus can persist in the semen of male survivors

Line 296 – 301: For case 8, what was the date of the positive Marburg test? What was the date of admission to the MTU?

Line 302 – 309: For case 9, it does not seem that she could have been infected by case 1, 4, or 8.

-Assuming an incubation period of 2-21 days and assuming her illness onset date is correct, she was exposed to Marburg virus 24 march – 6 april

-Case 1 died March 1

-Case 4 died March 16

-Case 8 was admitted to the hospital on March 21 and died April 10. Assuming case 8 was admitted to the MTU on March 21 and isolated, the Case 9 (mother of case 8) should not have had any contact with case 8 after he was admitted to the MTU

7. PLOS authors have the option to publish the peer review history of their article (what does this mean?). If published, this will include your full peer review and any attached files.

Reviewer #1: No

Reviewer #2: No

---

## [Author Response · Author response to Decision Letter 1]

23 Jul 2024

We would like to express our gratitude to the editor and reviewers for their valuable feedback on our manuscript titled "Epidemiological description of Marburg virus disease outbreak in Kagera region, Northwestern Tanzania". We have taken into account all of their suggestions and believe that our manuscript has been greatly improved as a result. We are confident that we have addressed all concerns and the revised manuscript is now ready to be re submitted

---

## [Decision Letter · Decision Letter 2]

19 Aug 2024

Epidemiological description of Marburg virus disease outbreak in Kagera region, Northwestern Tanzania

PONE-D-24-07742R2

Dear Dr. Mrema,

We’re pleased to inform you that your manuscript has been judged scientifically suitable for publication and will be formally accepted for publication once it meets all outstanding technical requirements.

Kind regards,

Masahiro Kajihara, PhD

Academic Editor

PLOS ONE

Additional Editor Comments (optional):

Reviewers' comments:

Reviewer's Responses to Questions

**Comments to the Author**

1. If the authors have adequately addressed your comments raised in a previous round of review and you feel that this manuscript is now acceptable for publication, you may indicate that here to bypass the “Comments to the Author” section, enter your conflict of interest statement in the “Confidential to Editor” section, and submit your "Accept" recommendation.

Reviewer #1: All comments have been addressed

Reviewer #2: All comments have been addressed

2. Is the manuscript technically sound, and do the data support the conclusions?

Reviewer #1: Yes

Reviewer #2: Yes

3. Has the statistical analysis been performed appropriately and rigorously? 

Reviewer #1: N/A

Reviewer #2: N/A

4. Have the authors made all data underlying the findings in their manuscript fully available?

Reviewer #1: Yes

Reviewer #2: Yes

5. Is the manuscript presented in an intelligible fashion and written in standard English?

Reviewer #1: Yes

Reviewer #2: Yes

6. Review Comments to the Author

Reviewer #1: The authors present an epidemiological description of the MVD outbreak in Tanzania. This is an interesting account of the outbreak response, the cases and circumstances surrounding each case.

All my comments have been addressed.

Reviewer #2: (No Response)

7. PLOS authors have the option to publish the peer review history of their article (what does this mean?). If published, this will include your full peer review and any attached files.

Reviewer #1: No

Reviewer #2: No

---

## [Editor Report · Acceptance letter]

26 Aug 2024

PONE-D-24-07742R2 

PLOS ONE

Dear Dr. Mrema, 

I'm pleased to inform you that your manuscript has been deemed suitable for publication in PLOS ONE. Congratulations! Your manuscript is now being handed over to our production team.

Kind regards, 

on behalf of

Dr. Masahiro Kajihara 

Academic Editor

PLOS ONE